# Is the TOAST Classification Suitable for Use in Personalized Medicine in Ischemic Stroke?

**DOI:** 10.3390/jpm12030496

**Published:** 2022-03-19

**Authors:** Sofie A. Simonsen, Anders S. West, Adam V. Heiberg, Frauke Wolfram, Poul J. Jennum, Helle K. Iversen

**Affiliations:** 1Clinical Stroke Research Unit, Department of Neurology, Rigshospitalet Glostrup, Valdemar Hansens Vej 1-23, 2600 Glostrup, Denmark; anders.sode.west@regionh.dk (A.S.W.); adam.vittrup.heiberg@regionh.dk (A.V.H.); helle.klingenberg.iversen@regionh.dk (H.K.I.); 2Department of Radiology, Rigshospitalet Glostrup, Valdemar Hansens Vej 1-23, 2600 Glostrup, Denmark; frauke.wolfram.02@regionh.dk; 3Danish Center for Sleep Medicine, Department of Neurophysiology, Rigshospitalet Glostrup, Valdemar Hansens Vej 1-23, 2600 Glostrup, Denmark; poul.joergen.jennum@regionh.dk; 4Faculty of Health and Medical Sciences, University of Copenhagen, 2100 Copenhagen, Denmark

**Keywords:** ischemic stroke, stroke, classification, magnetic resonance imaging, large vessel disease, small vessel disease, personalized medicine

## Abstract

Pathophysiologic classification of ischemic stroke is essential to a personalized approach to stroke treatment. The Trial of Org 101072 in Acute Stroke Treatment (TOAST) classification is the most frequently used tool to classify index ischemic strokes. We aimed to assess presence of small and large vessel disease markers across the TOAST groups. In an observational study, 99 ischemic stroke patients were consecutively included and classified according to TOAST. The assessment was supplemented with cerebral small vessel disease (SVD) score, based on Magnetic Resonance Imaging (MRI), and tests for carotid atherosclerosis, ankle–brachial index (ABI), estimated glomerular filtration rate (eGFR), and peripheral reactive hyperemia index (RHI). Markers of small and large vessel disease were present in all TOAST groups. Carotid stenosis and atrial fibrillation were associated with their respective TOAST groups (*p* = 0.023 and *p* < 0.001, respectively). We found no association between the SVD score and the small vessel occlusion TOAST group (*p* = 0.59), and carotid atherosclerosis (*p* = 0.35), RHI (*p* = 0.39), ABI (*p* = 0.20), and eGFR (*p* = 0.79) were not associated with TOAST groups. The TOAST classification does not provide differential information on the pathophysiologies of the ischemic stroke. An operational classification that contains quantification of each vascular pathophysiology in the individual patient is pivotal for future research and development of personalized medicine.

## 1. Introduction

Personalized medicine has gained more and more interest and has now come to stroke treatment [1]. Secondary preventive treatment is guided by the underlying vascular pathophysiology for ischemic stroke, e.g., direct oral anticoagulants for cardioembolic strokes, and initial dual platelet treatment for large vessel occlusion strokes [2].

The most frequently used method to classify the pathophysiology of the index ischemic stroke is the Trial of Org 101072 in Acute Stroke Treatment (TOAST) classification. TOAST consists of the following five groups: large artery atherosclerosis (LAA, ≥50% stenosis), cardioembolism (CE), small vessel occlusion (SVO), other determined etiology (OD), and undetermined etiology (UD) [3]. More than one third of patients are assigned to the UD group, consisting of patients with (1) multifactorial etiology, (2) inconclusive, or (3) inadequate examination [4]. Even though TOAST is a classification of the cause of the index stroke and offers only limited information about the presence of other vascular pathophysiologies relevant to stroke, TOAST is the most often used tool to stratify stroke patients in basic and clinical research. 

Peripheral artery disease measured as reduced ankle–brachial index (ABI) is found in up to half of all stroke patients [5] and is a risk factor for stroke recurrence [6]. Low ABI is related to intra- and extracranial stenosis as well as silent cerebral small vessel disease [7,8]. Endothelial dysfunction (ED), measured as reduced peripheral reactive hyperemia index (RHI), and renal insufficiency, measured as reduced estimated glomerular filtration rate (eGFR), are both more prevalent in stroke patients than in healthy individuals [9,10].

In 2013, a new precise and detailed method for describing cerebral small vessel disease (SVD) was proposed, and a total SVD score was defined. This was done as more than 50 different terms for SVD had been used until 2013 [11,12].

None of the mentioned markers are included in any stroke classification [13], and thereby potential modifiable risk factors are not identified and treated. 

The aim of the present study was first to describe the pattern of standard vascular markers in a cohort of stroke patients and second to test how additional markers of cerebral SVD and peripheral large and small vessel disease are distributed across TOAST groups. We hoped that additional markers could add valuable information to the TOAST classification, reduce the numbers of patients with undetermined etiology, and thereby improve the personalized medicine for the individual ischemic stroke patient. 

## 2. Materials and Methods

### 2.1. Design

We conducted an observational study of stroke patients over 18 years, independent of ethnicity, admitted to the stroke unit at the Department of Neurology, Rigshospitalet, Glostrup, Copenhagen University Hospital, Denmark, from May 2015 to August 2016 (clinicaltrials.gov: NCT02111408). 

Only patients with clinically verified ischemic stroke were included in this paper. Patients without the cognitive abilities to give consent and patients with other known brain diseases or short (months) life expectancies were excluded. 

All patients gave written informed consent. The study was approved by the Ethics Committee (H-2-2013-091) and the data-protecting agency (GLO-2013-18; IT suite nr. 02385). 

### 2.2. Examinations

The patients were examined within seven days from their index stroke. Based on a standard workup, the index stroke was classified according to the TOAST criteria. The standard workup included medical history, registration of medication and functional status (modified Rankin Scale, mRS), neurological examination including stroke severity (National Institutes of Health Stroke Scale, NIHSS), computed tomography (CT) or magnetic resonance imaging (MRI) of the brain (described below), chest X-ray, electrocardiogram, carotid ultrasound (described below), routine blood samples (including eGFR [14]), blood pressure, body mass index, and at least 48-h cardiac telemetry. Some patients also underwent echocardiography, CT or MR angiography, and extended blood examination if the basic examinations could not reveal an etiology for the stroke. The standard work-up was supplemented with assessment of peripheral large vessel disease, examined with the ankle–brachial index (ABI) and endothelial dysfunction, examined with the reactive hyperemia index (RHI), both described below. 

### 2.3. Magnetic Resonance Imaging

Patients without contraindications such as magnetic implants and severe claustrophobia were offered an MRI scan. MRI was performed with a Siemens Avanto 1.5 T scanner with a standard protocol including sagittal T2, axial T2, axial fluid attenuation inversion recovery, 3D T1, susceptibility-weighted imaging, and diffusion-weighted imaging. 

A blinded neuroradiologist provided a description of the intracerebral lesion(s) according to the Standards for Reporting Vascular Changes on Neuroimaging (STRIVE) [11]. The total SVD score was calculated with one point for each of the following: lacuna, deep and periventricular white matter hyperintensity (Fazekas score 2–3), microbleeds, and enlarged perivascular spaces (EPV) in the basal ganglia [12].

### 2.4. Carotid Ultrasound

Carotid ultrasound was performed as a part of the standard stroke work-up with a LOGIQ E9 (EG Healthcare). The presence of internal carotid artery stenosis was graded as 50–69%, 70–99%, or total occlusion according to international guidelines. Atherosclerosis was graded as no, mild, moderate, or severe [15].

### 2.5. Ankle–Brachial Index

The ABI was measured by a handheld doppler (Huntleigh Dopplex^®^ D900, Cardiff, Wales, United Kingdom) and a hand-operated blood pressure monitor (Welch Allyn®, Skaneateles Falls, NY, USA). After a 5-min supine rest, the blood pressure cuff was placed just above the ankle, and the systolic blood pressure in the dorsal pedis artery or posterior tibial artery was measured. Brachial systolic pressure was measured in each arm just before the ankle pressure with an automatic monitor (Omron M5-I, Hoofddorp, Netherlands). In patients where the automatic monitor had difficulties with measurements, the blood pressure was repeated until stable and the rest period extended. ABI was calculated as the lowest ankle pressure divided by the highest arm pressure [16]. An ABI between 0.9 and 1.4 was interpreted as normal [17].

### 2.6. Reactive Hyperemia Index

Endothelial function was assessed with the non-invasive EndoPAT™ (Itamar Medical Ltd., Caesarea, Israel) [18]. The EndoPAT™ measures bilateral beat-to-beat pulse wave amplitude in the index finger. A blood pressure cuff on the non-dominant arm was used to induce ischemia at a pressure of 30 mmHg above systolic pressure or at least 200 mmHg. The examination consisted of 6-min baseline, 5-min ischemia, and 3-min reactive hyperemia in a calm and temperature-controlled room of 21–24 °C with dimmed light. The automatic computerized RHI was used, and values below 1.67 were interpreted as pathological [19]. RHI is the ratio of the post-to-pre occlusion amplitude of the tested arm, divided by the post-to-pre occlusion amplitude of the control arm.

### 2.7. Statistical Analyses

Categorical data were presented as numbers and percentages and continuous data as mean and standard deviation (SD) or median and range (min–max) if not normally distributed. The SVD score was treated as a categorical variable.

Association between TOAST groups was tested with the Chi square or Fisher’s exact test for categorical data, as relevant, and analysis of variance and Kruskal–Wallis for continuous data with normal and non-normal distributions, respectively. Data were log-transformed to achieve normal distribution when necessary. Sex, but not age, was included as a confounder because only the distribution of sex differed among TOAST groups (see Results).

*P* values below 0.05 were considered significant. Statistical analyses were performed using RStudio version 1.0.136 (Boston, MA, USA). 

## 3. Results

Ninety-nine stroke patients were consecutively included within a median of 2 days from the index stroke (range 0–7 days). In total, 423 met the inclusion criteria, 50 patients refused to participate, 156 patients were discharged on the day where the diagnosis was certain, and 118 patients could not be included as we did not have the capacity to include more than one patient at a time as this paper presents data from a larger project with several other measurements (clinicaltrials.gov: NCT02111408). If more than one potential patient was admitted at the same day, the patient first admitted was included.

Median age was 68 years (range 36–88), 55 (56%) were men, and median NIHSS was 2 (range 0–16). MRI was performed in 89 patients, of whom 78 patients had acute ischemic lesions. One scan was impossible to assess due to poor quality, two patients had severe white matter lesions that masked an acute lesion, and eight patients had MR negative infarcts. Patient characteristics and vascular parameters for all patients and by TOAST groups are presented in Table 1. The UD group (39 of 99 patients) had 16 (41.0%) patients with ≥2 possible etiologies, 14 (35.9%) with incomplete examination, and 9 (23.1%) with inconclusive examination. The 16 patients with two or more possible etiologies had the following distribution: seven with LAA and CE, four with CE and SVO, three with LAA and SVO, one with SVO and OD, and one with LAA, CE, and SVO. 

### Vascular Parameters Related to TOAST Groups

Cerebral small vessel disease, expressed as the SVD score, was present in all TOAST groups (Figure 1), except the two patients with OD etiology (internal carotid artery dissection), who both had an SVD score of zero. We found no association between SVD score and TOAST classification (*p* = 0.59), though 28% and 37.2 % of the SVO and UD group had a SVD score of three or more, while this only accounted for 7.1% and 15.4% of the LAA and CE groups, and none had an SVD score of four. The SVO group had five (20%) patients with an SVD score of zero, as the acute lacunar infarct is not included in the SVD score.

Carotid stenosis ≥ 50% was associated with TOAST classification (*p* = 0.023) because these patients per definition belong to the LAA group, if not the UD. However, carotid stenosis was found across all TOAST groups (Table 1). Patients in the CE, SVO, and OD groups had stenosis on the non-relevant side. Carotid atherosclerosis was present in all TOAST groups (*p* = 0.35), but five (47%) of the patients in the LAA had severe atherosclerosis compared to only two (7%) in the SVO group (Figure 2). The two patients in the OD group both had mild carotid atherosclerosis.

In the CE group, eleven (68.8%) patients had atrial fibrillation (AF), two (12.5%) dilated cardiomyopathy, one (6.3%) congestive heart failure, one (6.3%) recent myocardial infarction, and one (6.3%) an akinetic left ventricular segment. AF was associated with the TOAST classification (*p* < 0.001) because these patients per definition belong to the CE group, if not to the UD group. 

Abnormal measurements of RHI (*p* = 0.55), ABI (*p* = 0.41), and eGFR (*p* = 0.86) were present in all TOAST groups (Figure 3, Figure 4 and Figure 5). Patients in the LAA group stood out regarding RHI, with only 2 (17%) patients with abnormal values compared to 31–39% in the other groups. Variances in RHI, ABI, and eGFR were equal across the TOAST groups (Table 1). 

## 4. Discussion

This observational study showed a distribution across TOAST groups comparable to previous studies [4]. We had expected the SVD score to be associated with the SVO group. Instead, we found cerebral SVD and carotid atherosclerosis as well as abnormal RHI, ABI, and eGFR values in patients across all TOAST groups. Thus, inclusion of additional vascular markers did not improve the TOAST classification for the individual patient. As expected, ipsilateral carotid stenosis ≥ 50% and atrial fibrillation were associated with the TOAST classification. 

A high SVD score has been tied to lacunar strokes in some [12,20] but not all studies [21]. Low RHI has been associated with SVO, CE, UD, and LAA groups [9,22]. The SVO group was least affected by abnormal ABI in one study [23], while another found no correlation between ABI and the TOAST classification [5]. The CE group was most affected by renal insufficiency in one study [24] and least in another [25]. Bilateral non-stenotic carotid atherosclerosis has been described in both cryptogenic stroke and in stroke in patients with atrial fibrillation [26,27]. These inconsistent results can be explained by the co-existing vascular pathologies across the TOAST groups that are not included in the classification because TOAST classifies the pathophysiology only of the index stroke. In addition, the LAA classification limits stenosis to ≥50% ipsilateral to the infarct, and the SVO classification allows only infarcts of 15 mm or smaller. A patient with a stenosis below 50% will not be classified as having large vessel disease, though this patient group recently has been shown to benefit from a different treatment than patients without stenosis [28].

Another limitation of the TOAST classification is the large UD group (39% in the present study). With a classification system that allows more than one possible etiology and sufficient examination of all patients, this group could be reduced to only nine patients (9%) in this study.

Several other classification systems have been developed with more well-defined groups and to provide both causative and phenotypic classifications: The Stop Stroke Study TOAST (SSS-TOAST) [29], the Causative Classification of Stroke System (CCS), a web-based causative and phenotypic version of SSS-TOAST [30], and the phenotypic ASCO score (A for atherosclerosis, S for small vessel disease, C for cardiac source, O for other cause) [31]. The ASCO score was later extended with a “D” for dissection [32].

The ASCO(D) classification uses leukoaraiosis, microbleeds, and EPV as markers for cerebral SVD, together with lacunar infarcts, but lacks a precise definition for the listed markers. SSS-TOAST and CCS are the only classification systems that allow lacunae up to 20 mm, like the newest guidelines [11].

CCS and ASCOD accept stenosis below 50%, and ASCO uses carotid plaque as a marker of large vessel disease. 

None of the existing classification methods are operational or include both cardioembolic sources, precise definition of objective measures of cerebral SVD including acceptance of infarcts sized up to 20 mm and any large artery pathophysiology, both stenosis and atherosclerosis [13].

The need for a classification system that includes all underlying pathophysiological mechanisms in the individual ischemic stroke patient has been raised in an Asia-specific context [33], but this need is general, especially in terms of research (e.g., into genetics and biomarkers), choice of treatments, and further development of personalized medicine [1,34]. It is known that patients with atrial fibrillation (AF) should be treated with DOAKs or vitamin K antagonists, while patients with large vessel disease benefit from anti platelet aggregation treatment and carotid endarterectomy if stenosis is present. However, some patients suffer from both AF and large vessel disease. 

In stroke research with a focus on genetics, biomarkers, and omics, etc., it is crucial to have a “fingerprint” of the presence and severity of the vascular pathophysiologies in the individual patient to be able to uncover basic mechanisms. This will provide knowledge for future studies tailoring the treatment according to the individual pattern of pathophysiology. 

### Strengths and Limitations

The main strength of this study is that we focus on whether the TOAST classification reflects the vascular profile of the individual patient and if adding further vascular tests than standard could improve the TOAST classification on the individual level.

Furthermore, we relate the relatively new total SVD score, carotid atherosclerosis, ABI, and RHI together with the standard stroke work-up to the TOAST classification. The patients were consecutively included but because of physical and cognitive disabilities, we could not include those who had severe strokes, and not all patients could participate in all examinations. This limits the generalizability to this patient group; however, we have no reason to think that these patients have fewer vascular pathophysiologies than the included patients. 

The sample size was relatively small, and it cannot be ruled out that ABI, RHI, and eGFR in a larger population could be related to the TOAST classification subgroups. However, this will not increase the usefulness of the TOAST classification on the individual level, as we were able to show that patients in all TOAST groups have co-existing vascular pathophysiologies. In addition, as seen graphically in Figure 1, it seems like the SVD score is higher in the SVO and UD group, and we cannot exclude that a larger sample size could show a statistically significant association.

## 5. Conclusions

Cerebral SVD were present across all TOAST groups and objective measures of large and small vessel disease (carotid atherosclerosis, ABI, RHI and eGFR) were not associated to specific TOAST groups. Only strokes in patients with ipsilateral carotid stenosis ≥ 50% and atrial fibrillation, were associated with TOAST.

A prerequisite for personalized medicine is knowledge of the pathophysiology in the individual patient and the TOAST classification does not provide this information. In stroke research with focus on genetics, biomarkers, and omics, etc., it is crucial to have a “fingerprint” of the presence and severity of the vascular pathophysiologies in the individual patient, to be able to uncover basic mechanisms. This will provide knowledge for future studies tailoring the treatment according to the individual pattern of pathophysiology. Therefore, we find a need for a new operational stroke classification including description and quantification of all vascular pathophysiologies. 

## Figures and Tables

**Figure 1 jpm-12-00496-f001:**
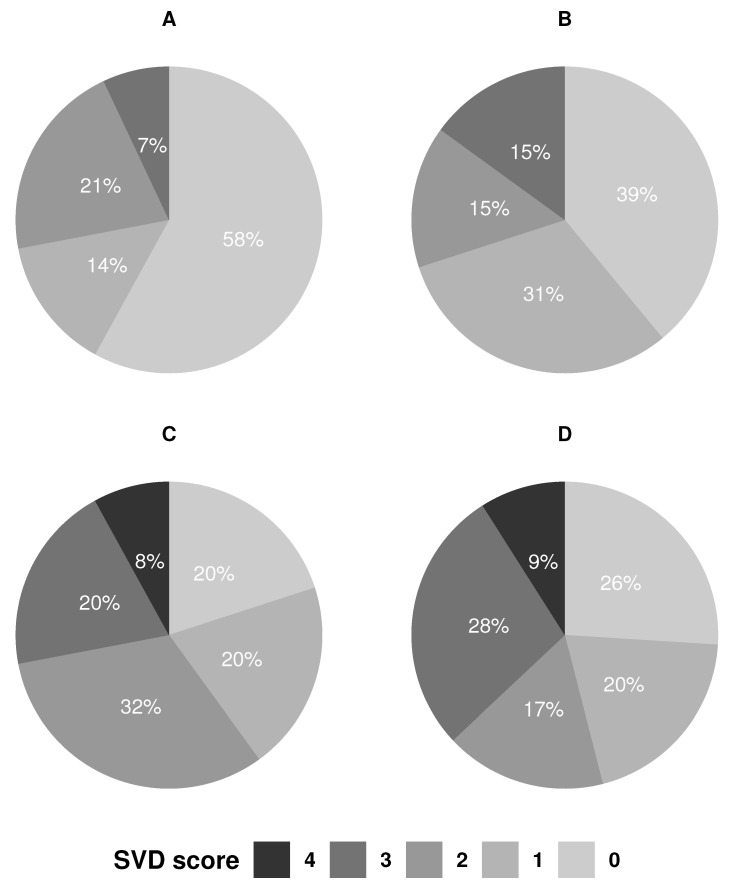
Distribution of the SVD score across TOAST groups. CE: cardioembolism; LAA: large artery atherosclerosis; SVO: small vessel occlusion; UD: undetermined etiology.

**Figure 2 jpm-12-00496-f002:**
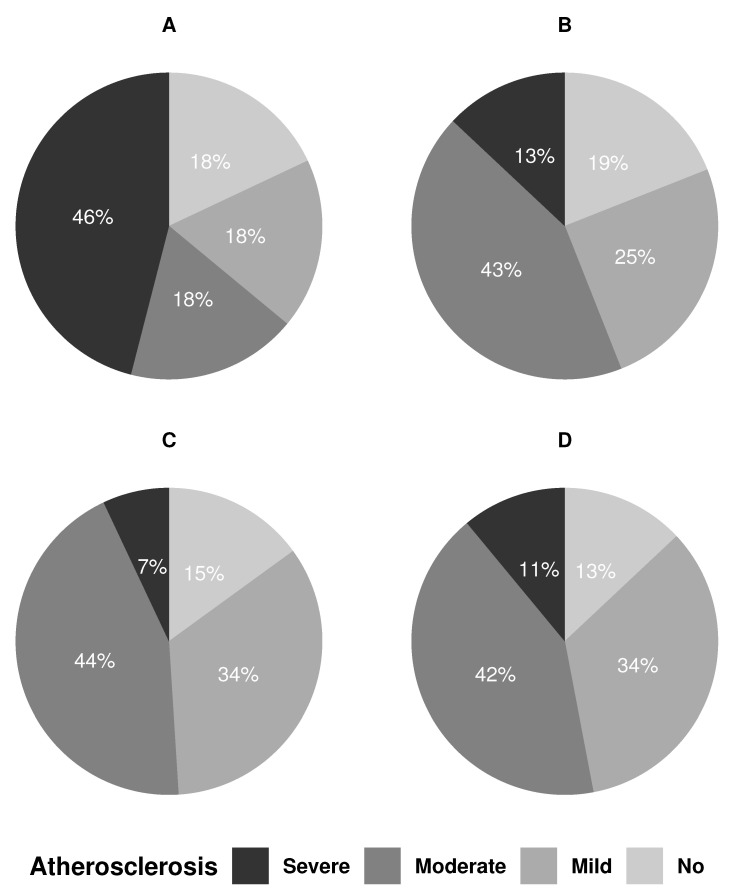
Distribution of internal carotid artery atherosclerosis across TOAST groups. CE: cardioembolism; LAA: large artery atherosclerosis; SVO: small vessel occlusion; UD: undetermined etiology.

**Figure 3 jpm-12-00496-f003:**
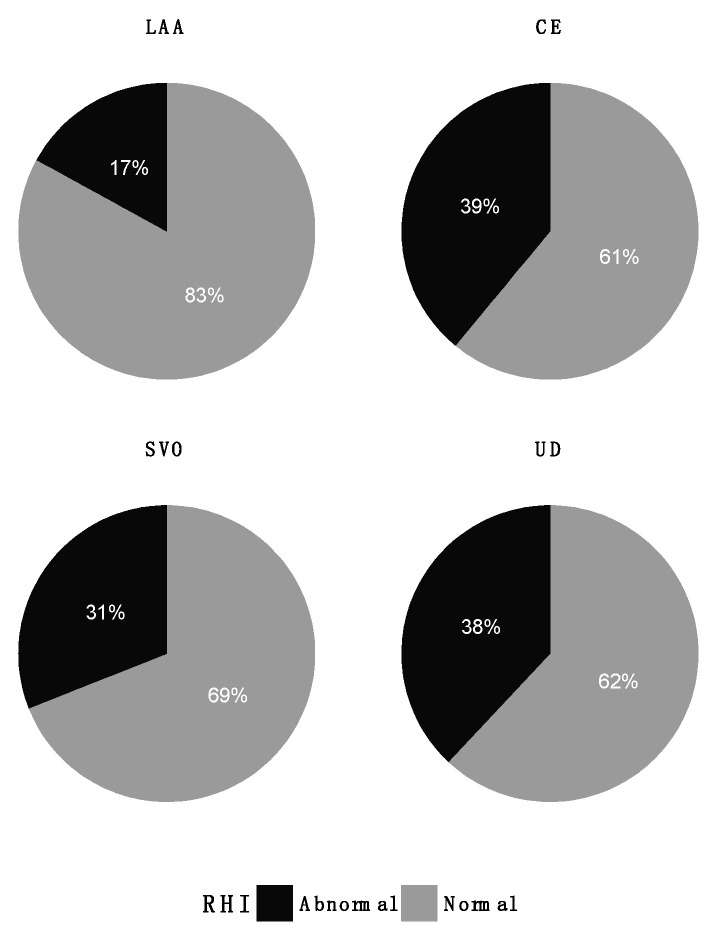
Distribution of abnormal RHI (<1.67) across TOAST groups. CE: cardioembolism; LAA: large artery atherosclerosis; SVO: small vessel occlusion; UD: undetermined etiology.

**Figure 4 jpm-12-00496-f004:**
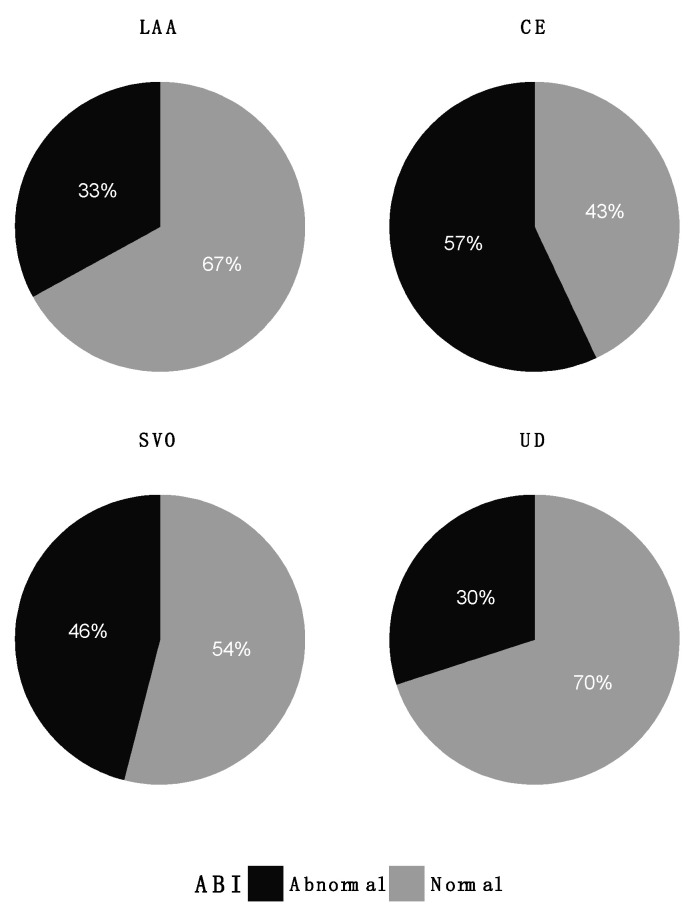
Distribution of abnormal ABI (<0.9 or >1.4) across TOAST groups. CE: cardioembolism; LAA: large artery atherosclerosis; SVO: small vessel occlusion; UD: undetermined etiology.

**Figure 5 jpm-12-00496-f005:**
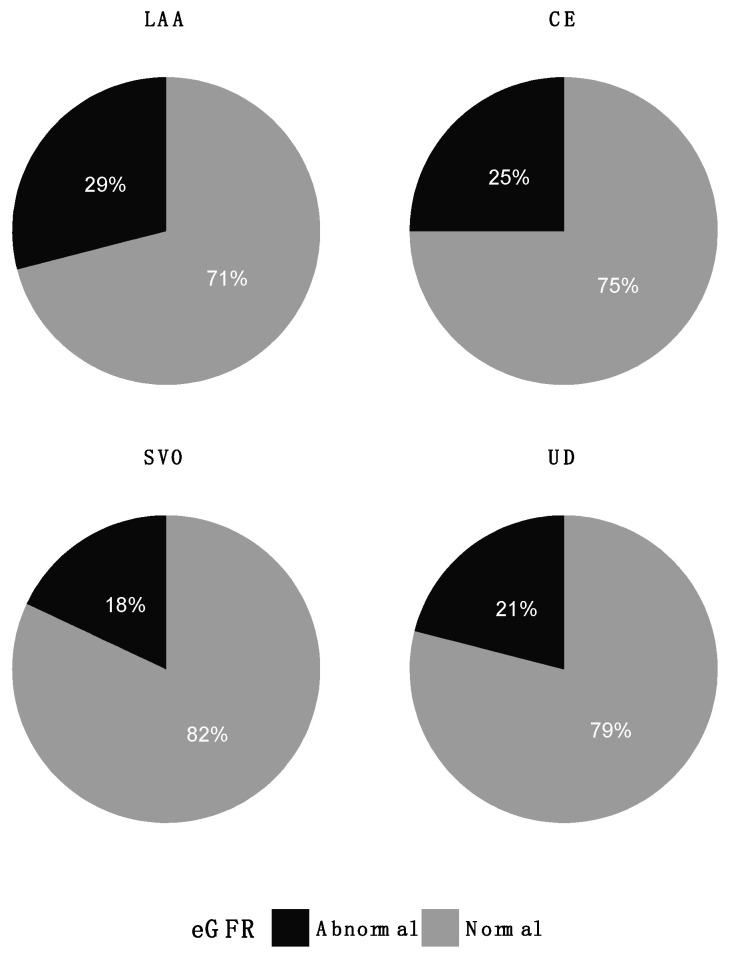
Distribution of abnormal eGFR (<60 mL/min/1.73 m^2^) across TOAST groups. CE: cardioembolism; LAA: large artery atherosclerosis; SVO: small vessel occlusion; UD: undetermined etiology.

**Table 1 jpm-12-00496-t001:** Patient characteristics and vascular parameters for all patients and by TOAST groups.

	All Patients	Large Artery Atherosclerosis	Cardio-Embolism	Small Vessel Occlusion	Other Determined Etiology	Undetermined Etiology	*p*-Value
	*n* = 99	*n* = 14	*n* = 16	*n* = 28	*n* = 2	*n* = 39	
Age, years	68 (36–88)	68 (45–81)	73 (36–83)	71 (47–88)	52 (51–52)	68 (41–85)	0.082
Sex							0.017
Men	55 (55.6)	5 (35.7)	9 (56.2)	11 (39.3)	1 (50.0)	29 (74.4)	
Woman	44 (44.4)	9 (64.3)	7 (43.8)	17 (60.7)	1 (50.0)	10 (25.6)	
mRS	2 (0–5)	2 (0–4)	1 (0–4)	2 (0–4)	2 (2–2)	2 (0–5)	0.52
NIHSS	2 (0–16)	2 (0–13)	1 (0–13)	2 (0–9)	2 (1–3)	2 (0–16)	0.39
Comorbidities							
Hyper-tension	59 (59.6)	6 (42.9)	11 (68.8)	17 (60.7)	1 (50.0)	24 (61.5)	0.67
Diabetes type 2	20 (20.2)	2 (14.3)	6 (37.5)	4 (14.3)	0 (0.0)	8 (20.5)	0.41
Hypercholesterolemia	60 (60.6)	10 (71.4)	9 (56.2)	19 (67.9)	1 (50.0)	21 (53.8)	0.68
Atrial fibrillation	18 (18.2)	0 (0.0)	11 (68.8)	0 (0.0)	0 (0.0)	8 (20.5)	<0.001
Former stroke or TIA	25 (25.3)	4 (28.6)	5 (31.2)	10 (35.7)	0 (0.0)	6 (15.4)	0.32
SVD score	*n* = 89	*n* = 14	*n* = 13	*n* = 25	*n* = 2	*n* = 35	0.59
0	29 (32.6)	8 (57.1)	5 (38.5)	5 (20.0)	2 (100.0)	9 (25.7)	
1	18 (20.2)	2 (14.3)	4 (30.8)	5 (20.0)	0 (0.0)	7 (20.0)	
2	19 (21.3)	3 (21.4)	2 (15.4)	8 (32.0)	0 (0.0)	6 (17.1)	
3	18 (20.2)	1 (7.1)	2 (15.4)	5 (20.0)	0 (0.0)	10 (28.6)	
4	5 (5.6)	0 (0.0)	0 (0.0)	2 (8.0)	0 (0.0)	3 (8.6)	
Carotid examination	*n* = 94	*n* = 11	*n* = 16	*n* = 27	*n* = 2	*n* = 38	
Carotid stenosis ≥ 50%	17 (17.2)	6 (54.5)	1 (6.3)	2 (7.4)	1 (50.0)	7 (18.4)	0.023
Endothelial dysfunction	*n* = 90	*n* = 12	*n* = 13	*n* = 26	*n* = 2	*n* = 37	
RHI	1.89 (0.82–3.81)	2.21 (1.37–3.81)	1.85 (1.28–2.42)	1.83 (0.92–3.80)	2.21 (1.86–2.55)	1.90 (0.82–3.35)	0.37
Peripheral arterial disease	*n* = 89	*n* = 12	*n* = 14	*n* = 24	*n* = 2	*n* = 37	
ABI, mean (SD)	0.93 (0.18)	0.99 (0.11)	0.88 (0.17)	0.89 (0.19)	0.81 (0.13)	0.97 (0.20)	0.20
Renal function	*n* = 99	*n* = 14	*n* = 16	*n* = 28	*n* = 2	*n* = 39	
eGFR, mL/min/1.73 m^2^	82 (6–116)	81 (39–116)	81 (43–102)	83 (6–102)	95 (80–109)	78 (18–105)	0.79

Categorical variables are presented as n (%) and continuous variables as median (range) if not otherwise stated. Comorbidities are defined as history or medical treatment for the diseases at the time of the index stroke. *P*-value is for association between TOAST groups; for specific tests, see Statistics. ABI: ankle–brachial index; eGFR: estimated glomerular filtration rate; mRS: modified Rankin scale; NIHSS: National Institutes of Health Stroke Scale; RHI: reactive hyperemia index; SVD: small vessel disease, TIA: transient ischemic attack.

## Data Availability

The data presented in this study are available upon request from the corresponding author. The data are not publicly available due to local data protecting rules.

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
