# Peer review of "Is the TOAST Classification Suitable for Use in Personalized Medicine in Ischemic Stroke?"

_jpm, 2022, doi:10.3390/jpm12030496_

Round 1

Reviewer 1 Report

The author present an evaluation of 99 stroke patients with regard to TOAST classification and some measures of large and small vessel disease. They conclude that imaging characteristics of small vessel disease and other markers of large and small vessel disease are present among all TOAST subgroups. According to the authors, TOAST criteria are insufficient in determining stroke etiology. Other subclassification systems of stroke etiology are discussed briefly. 

There are several limitations to the study, some of which have been mentioned by the authors.

687 patients were screened but only 99 were included. Unfortunately, 118 patients were excluded due to limited inclusion resources. The exclusion of more than three quarters of patients that were candidates for inclusion might have introduced considerable bias. The study size is rather limited which limits statistical power. There has obviously been no prior statistical power calculation to estimate the necessary sample size. Severe strokes and demented patients were excluded which limits generalizability of their findings.

Included strokes were “clinically verified”. This seems to imply that some of them had no imaging correlates. If so, how many of them?

As only few patients underwent echocardiography und ECG monitoring was limited to 48 hours, some cases of cardioembolic etiology might have been missed.

The reactive hyperemia index has been used in relatively few acute stroke studies. Although two recent studies found an association of RHI with incident stroke or cryptogenic stroke in men or patients > 41 years, it is possible that acute stroke introduces further changes in RHI. Longitudinal studies would clarify this point.

We concur with the authors that the large proportion of undetermined stroke etiology according to TOAST is dissatisfying. The authors suggest that a description of all vascular pathologies is needed to provide personalized medicine. For the time being, there has been no evidence that such an all-encompassing approach is appropriate to tailor treatment decisions or even to improve prognosis. The suggested “quantification of each vascular pathophysiology” has to be substantiated. Will there be a score, several scores / grading systems, or simply descriptive terms? Anyway, any of this has to be translated into solid treatment decisions. According to which criteria? Do the authors think that further tests will help? Thus, the authors should mention which additional examinations they think might help in classifying stroke subtypes more correctly in the future - for example, blood biomarkers? Or do they suggest to abandon subtyping altogether?

Reviewer 2 Report

Since an 80 year old study subject with a serum creatinine of 1.0 mg/dl has an eGFR< 60 ml/min while a 60 year old study subject with a serum creatinine of 1.0 mg/dl has an eGFR > 60 ml/min, you may want to add a comment about this strict cutoff for a population with a mean age of ~ 70 years of age.

Reviewer 3 Report

This observational study described the vascular pathophysiological pattern in a cohort of stroke patients and investigated how cerebral SVD and markers of peripheral large and small vessel disease are distributed across TOAST groups. The study found no association between the SVD score and the small vessel occlusion TOAST group (p=0.59), and carotid atherosclerosis (p=0.35), reactive hyperemia index (p=0.39), ankle–brachial index (p=0.20), and estimated glomerular filtration rate (p=0.79) were not associated with TOAST groups. Here are my suggestions:

  1. The process of carotid ultrasound needs to be briefly described. Inter- and intra-rater reliability extracted from previous studies or the subjects of this study should be presented. The grade of atherosclerosis needs to be clearly stated along with the stenosis percentage range. Was the carotid artery stenosis (i.e. ≥50%) PSV 125 cm/s or greater?
  2. Ankle-brachial index is calculated based on BP measurement. Efforts to reduce BP fluctuations should be described. For example, ABI measurements in patients with atrial fibrillation and sinus tachycardia are not reliable.
  3. A flow chart for inclusion and exclusion of patients is required.
  4. Large scale studies showed the SVD score was associated with SVO subtype. ABI also has a significantly higher prevalence in patients with LAA or atrial fibrillation. However, this study did not show a statistically significant difference between the stroke subtypes for these. Potential reasons for this appear to be insufficient sample size and unequal number of subjects between groups. The authors require additional statistical methods to overcome this.
  5. It is recommended to investigate whether the distribution of peripheral vascular markers differs between TOAST and more sophisticated classifications (SSS-TOAST, CCS, or ASCO).
  6. Peripheral vascular markers are independently associated with stroke prognosis. However, the treatment for stroke patients with peripheral vascular disorder is not well established. This study still does not provide conclusive evidence that these markers are helpful for treatment at the precision medicine level.

Round 2

Reviewer 3 Report

Accept in present form